# Adsorption of the Tartrate Ions in the Hydroxyapatite/Aqueous Solution of NaCl System

**DOI:** 10.3390/ma14113039

**Published:** 2021-06-03

**Authors:** Władysław Janusz, Ewa Skwarek

**Affiliations:** Department of Radiochemistry and Environmental Chemistry, Faculty of Chemistry, Maria Curie-Skłodowska University, 3 Maria Curie-Skłodowska Square, 20-031 Lublin, Poland; wladyslaw.janusz@poczta.umcs.lublin.pl

**Keywords:** hydroxyapatite structure, adsorption ions, tartrate ions

## Abstract

The research on the interaction of tartrate ions with the surface of hydroskyapatite was presented, including the measurements of the kinetics of tartrate ion adsorption and tartrate ion adsorption as a function of pH. The adsorption of tartrate ions was calculated from the loss of tartrate concentration in the solution as measured by a radioisotope method using C-14 labeled tartaric acid. In order to explain the mechanism of interaction of tartrate ions with hydroxyapatite, supplementary measurements were carried out, i.e., potentiometric measurements of the balance of released/consumed ions in the hydroxyapatite/electrolyte solution system, zeta potential measurements, FTIR spectrophotometric measurements and the hydroxyapatite crystal structure and particle size distribution were characterized. It was found that the adsorption of tartrate ions occurs as a result of the exchange of these ions with hydroxyl, phosphate and carbonate ions. Replacing the ions with the abovementioned tartrate ions leads to the appearance of a negative charge on the surface of the hydroxapatite. On the basis of XRD study and particle size distribution, a decrease in the size of crystallites and the diameter of hydroxyapatite particles in contact with a solution of 0.001 mol/dm^3^ of tartaric acid was found.

## 1. Introduction

Teeth and bones are a natural composite of organic and inorganic substances in which hydroxyapatite is the inorganic component. Its content in teeth is largest in the mature enamel (it amounts to 96% by weight), whereas in the dentin it is similar to that in bone (~70% by weight) [1,2]. The enamel surface is composed of 30 nm × 60 nm hydroxyapatite crystals with a length of several µm. During meals consumption, teeth undergo physical and chemical processes. In chemical processes, the tooth surface (enamel) is subjected to interactions with food components or the metabolism of acid bacteria, as a result of which its erosion may occur [1]. Many beverages contain organic carboxylic acids which due to dissociation release hydrogen ions and which then take part in the dissolution reactions of hydroxyapatite. On the other hand, the anions originating from dissociation of polycarboxylic acid are characterized by having calcium ion complexing properties, which additionally contribute to dental erosion [3]. Organic acids interact with hydroxyapatite through the erosion mechanisms, for example, with citric acid whose interactions with hydroxyapatite have been discussed in many papers [4,5,6,7,8,9]. Like citric acid, tartaric acid has the first- and second-degree dissociation constants as well as calcium ion complexing constants [3]. However, unlike tartrate ion with calcium ions, it forms a relatively sparingly soluble precipitate [10]. In some fruits (e.g., grapes, tamarind or garcinia) 100 g of tartaric acid is present in the concentrations of several mg/100 g, while in the others, e.g., (apples, oranges, strawberries) in tenths of mg/100 g [11]. It can be also found in cocktails, various drinks made of these fruits. Therefore, tartaric acid can cause dental erosion when eating fruit or drinking beverages [12]. Studies were carried out also on the interactions of tartaric acid with hydroxyapatite in order to use it as an etching agent and a component of self-adhesive primers in adhesive dentistry [13].

Research on the interactions of low-molecular organic (oxalic, malic and citric) acids on the samples of hydroxyapatite with varying degrees of crystallinity was also carried out regarding their interactions in the soil systems [14]. If was proved that the adsorption of anions of these acids decreases with the increasing pH and that the adsorption isotherms are better fitted to the Freudlich isotherm. Oxalic acid has also been shown to have stronger adsorption affinity for hydroxyapatite than malic and citric acids. The citrate ion adsorption studies presented in the paper by Vega et al. proved that the adsorption process involves the interactions of the citrate molecule with two hydroxyapatite adsorption sites [15]. The different adsorption affinity of citrate acid for hydroxyapatite crystal walls affects the nucleation and growth of hydroxyapatite crystals so that hydroxyapatite with different morphology can be obtained [16]. 

In this paper, the changes of the adsorption vs. time and dependence of adsorption of tartrate ions as a function of pH on the hydroxyapatite were studied using 14C radiolabeled tartaric acid from the solutions of tartaric acid of the initial concentration from 0.000001 mol/dm^3^ to 0.001 mol/dm^3^ and the pH range from 6 to 11 in the 0.001 mol/dm^3^ aqueous solution of NaCl. The results of the tartaric acid adsorption measurements were supplemented with the results of FTIR spectra, potentiometric titration of HAp suspensions and electrophoretic measurements of zeta potential and particle size analysis using static light scattering.

## 2. Materials and Methods

The solutions used in this study were prepared from the following reagents: DL-tartaric acid (POCh, Gliwice, Poland, analytical reagent grade), NaCl (Sigma Aldrich product, analytical reagent grade, Saint Louis, MO, USA), NaOH (Sigma Aldrich product, analytical reagent grade), HCl (Sigma Aldrich product, analytical reagent grade), Tartaric acid DL-[1,4,14C] (American Radiolabeled Chemical, Inc., St. Louis, MO, USA) The solutions were prepared by dissolving the above-mentioned reagents in de-ionized water from Millipore Milli-Q system (Burlington, MA, USA). 

Synthetic hydroxyapatite obtained by the wet method by Ca(OH)_2_ reaction with H_3_PO_4_ was used in the research carried out in this study [17]. After synthesis, the filter cake was dispersed in distilled water and washed several times with redistilled water to remove unreacted substances until constant conductivity (about 10 µS/cm) was achieved.

The crystal structure of hydroxyapatite and hydroxyapatite with adsorbed tartaric acid samples were studied by powder X-ray diffraction (PXRD) by means of an X-ray diffractometer Empyrean produced PANalytical (Malvern, Worcestershire, UK). The specific surface area of the hydroxyapatite samples and their porosity were calculated using the BET or BJH method, respectively, based on complete nitrogen adsorption/desorption isotherms obtained with the (ASAP2405 apparatus (Accelerated Surface Area and Porosimetry, Micrometrics Instruments, Co., Norcross, GA, USA)

The adsorption of tartrate ions on the hydroxyapatite was calculated on the basis of the uptake of these ions from the solution after introducing into it from the hydroxyapatite sample. Changes in the concentration of tartrate ions were determined by the isotope tracing method using 14C-labeled tartaric acid. Tartaric acid adsorption studies were carried out in a Teflon vessel. The pH of the suspension was controlled using a PHM 240 pH meter, Radiometer with glass (pHG201-8) and calomel electrodes (REF-451) attached. In order to eliminate the influence of CO_2_ on the pH measurements and its adsorption on the hydroxyapatite, nitrogen purified from CO_2_ flowed through the vessel. The adsorption kinetics were measured for two initial concentrations of tartaric acid, i.e., 0.000001 and 0.001 mol/dm^3^. On the other hand, the measurements of tartaric acid adsorption as a function of pH were carried out in the pH range from 6–11.5 and for the following initial concentrations of tartaric acid: 0.000001, 0.00001, 0.0001 and 0.001 mol/dm^3^. In addition to the adsorption measurements, potentiometric titration of the hydrosyapatite suspension in the presence and absence of tartaric acid was carried out in order to determine the bound or released H^+^ ions in the system by adsorbing tartaric acid. The FTIR ATR spectra of the hydroxyapatite samples in the range 4000–400 cm^−1^ were recorded using FTIR Nicolet 8700A Thermo Scientific (Waltham, MA, USA).

The zeta potential of hydroxyapatite was calculated from electrophoretic mobility of particles determined using Zetasizer 3000 (Malvern, Malvern, UK). Prior the electrophoretic measurements, suspension of 100 ppm hydroxyapatite concentration was dispersed by ultrasound waves. To ensure a constant ionic strength, the adsorption measurement and electrophoretic measurements were carried out in 0.001 mol/dm^3^ NaCl as the basic electrolyte.

## 3. Results and Discussion

The research on the interactions of low-molecular organic acids with hydroxyapatite, on one hand, must take into account the reactions of hydroxyapatite dissolution and the formation of complexes of organic acid ions with calcium ions in the solution as well as possible precipitation of a new phase in the case of exceeding the solubility of the calcium salt of the carboxylic acid. In addition to these bulk reactions, adsorption of carboxylic acid anions may occur on the hydroxyapatite surface. The research results presented so far have shown that for the small initial concentrations of oxalic acid (e.g., 0.000001 mol·dm^−3^), anion adsorption is fast whereas for the initial concentration 0.001 mol·dm^−3^, in the first stage, fast adsorption is followed by a slow adsorption process. These studies would show that the adsorption of oxalate anions occurs through the exchange of ions partly with hydroxyl groups and partly with hydroxyapatite phosphate ones [18]. 

The solubility diagram of hydroxyapatite in the aqueous solutions indicates that the minimum solubility of hydroxyapatite occurs at pH = 12. Below this value, the solubility increases with the decreases in pH [3,19,20,21,22]. As mentioned above, the presence of tartrate ions in the system will affect calcium ions, and it can also lead to the formation of calcium tartrate precipitate. To determine the conditions of hydroxyapatite transformation into calcium tartrate, hydroxyapatite solubility diagrams were prepared based on the hydroxyapatite solubility product pKs0 (Ca_5_(PO_4_)_3_OH) = 57.43 [23] dissociation constants for phosphoric acid pK_a1_ = 2.16, pK_a2_ = 7.21 and pK_a3_ = 12.32 [24], stability constants of calcium phosphate complexes pK (CaH_2_PO_4^+^_) = −0.84, pK (CaHPO_4_) = −2.55 and pK (CaPO_4^−^_) = 5.54 [25], calcium hydoroxocomplex instability pK (CaOH) = 1.2 [26]. The calcium tartrate solubility diagram was prepared based on the calcium tartrate solubility product pK_So_ (CaC_6_O_4_H_4_) = 6.11 [10] instability of calcium tartrate complexes constants pK (CaTar(_aq_)) = 2.01 and pK (CaHTar^+^) = 1.16 [27], tartaric acid dissociation constants pK_a1_ = 3.03 and pK_a2_ = 4.37 [24]. 

Based on the solubility diagrams of hydroxyapatite in an aqueous solution and calcium tartrate in water, the relationship of the negative logarithm of the concentration of calcium species in water was coupled for both compounds as a function of pH, Figure 1. 

The comparison of the relationships shown in Figure 1 indicates that the curves intersect at pH = 5.45. Below this value, calcium tartrate is the more sparingly soluble precipitate, in the presence of hydroxyapatite. Greenwald has shown that in complex systems, e.g., in the hydroxyapatite suspension containing tartaric acid ions, the solubility of the hydroxyapatite increases [28]. 

As mentioned above, calcium ions form with the tartar ions two complexes CaTar_(aq)_ and CaHTar^+^ which will be present in aqueous solutions. The example of the share of ionic forms in a solution containing 0.001 mol/dm^3^ Ca^2+^ ions and 0.001 mol/dm^3^ tartrate ions as a function of pH is shown in Figure 2, which presents changes in the share of phosphate and tartrate species in the pH function. As can be seen from the comparison of the forms after dissociation of the first-degree acids, phosphoric acid is a stronger acid, but in the second dissociation degree, the tartrate anion turns out to be a stronger acid than the dihydrogen phosphate anion. Due to the solubility of hydroxyapatite, the pH range of the adsorption tests is from 6 to 11 (in Figure 2, it is marked with a rectangle). In this pH range, tartrate ions will be present in the solution in contact with hydroxyapatite, and up to pH = 7.2, dihydrogen phosphate ions will dominate and above this pH value hydrogen phosphate ions.

### 3.1. Kinetics of Tartrate Ions Adsorption Process

The previous studies on the adsorption of polycarboxylic acids on hydroxyapatite showed that adsorption can occur as a result of interaction (coordination) with calcium ions [14], due to ion exchange with hydroxyapatite anions, i.e., with hydroxyl or phosphate ions [8] and as a result of hydroxyapatite dissolution and recrystallization processes [18]. Ion adsorption kinetics at the boundary solid electrolyte solution can be described by various models [29]. To describe the kinetics of ion adsorption on the hydroxyapatite/electrolyte solution boundary, we applied the Elovic equation, intraparticle diffusion, pseudo first order and pseudo second equations [30]. In the case of oxalic acid adsorption on hydroxyapatite from a solution with a small initial concentration (0.000001 mol/dm^3^), a good match of the adsorption kinetics was obtained using the pseudo second order equation. At the initial concentration of 0.001 mol/dm^3^, the adsorption kinetics is complex, and a good fit was achieved using the multi-potential model [29,31]:(1)$$\frac{c}{c_{0}}=A_{0}+\sum_{i=1}^{n}A_{i}\exp\left(-k_{i}t\right)\;\;\;where\;\;A_{0}+\sum_{i=1}^{n}A_{i}=1\;$$
(2)$$\frac{a}{a_{eq}}=1-\frac{\sum_{i+1}^{n}A_{i}\exp\left(-k,t\right)}{\sum_{i+1}^{n}A_{i}}=1-\frac{\sum_{i=1}^{n}A_{i}\exp\left(-k,t\right)}{1-A_{0}}=1-\sum_{i=1}^{n}\frac{A_{i}}{1-A_{0}}\exp\left(-k_{i}t\right)\;$$
where
*a*—the adsorption.*a_eq_*—the equilibrium adsorption.*c*_0_—the initial concentration.*A*_0_—the relative equilibrium concentration.*A_i_*—describes the part of the *i*-th process characterized by the coefficient *k_i_*.

The results of the study on the kinetics of adsorption of tartar ions on hydroxyapatite from a solution with an initial concentration of 0.000001 mol/dm^3^ and the kinetics of pH exchange for this system are shown in Figure 3. Similar relationships regarding the tartrate ion adsorption from a solution with 0.001 mol/dm^3^ tartrate ion are presented in Figure 4. Adsorption of tartrate ions from a solution with 0.00001 mol/dm^3^ ion concentration reaches equilibrium after one minute. In Figure 3, the dashed line with short segments shows the fit of the pseudo first order model and the line with longer segments the fit of the pseudo second order model. As you can see, the pseudo second order equation describes the adsorption kinetics better than the pseudo first order equation. The kinetics of tartar ion adsorption on hydroxyapatite from a solution with an initial concentration of 0.001 mol/dm^3^ of tartrate ions and the changes in pH are presented in Figure 4. In the course of tartrate ion adsorption as a function of time, there can be distinguished three stages of adsorption kinetic: in the first stage quick adsorption is in a few minutes followed by stage II in which adsorption grows slowly and in stage III the adsorption increases again, and the equilibrium is established. At the same time, there is an increase in pH at the observed time of 200 min adsorption at pH = 6.3–6.9. The increase in adsorption may be due to the dissolution of hydroxyapatite and the pursuit of the system to a minimum of solubility as well as a result of adsorption involving the exchange of tartrate to hydroxyl ions. Figure 4 presents the results of matching the adsorption kinetics with the pseudo first order and pseudo second order equations as well as using a multi-potential equation. This last equation describes the course of adsorption best, which may indicate a complex process of adsorption of tartrate ions on hydroxyapatite. The initial stage of tartrate ion adsorption can be associated with phosphate and hydroxyl ion exchange reactions as indicated by an increase in pH. The slow adsorption step can be associated with the dissolution and recrystallization of hydroxyapatite or/diffusion in type II hydroxyapatite crystal lattice channels having a diameter of 0.45 nm [32] in which phosphate ions can be exchanged with other anions [33].

The dependence of tartrate ion adsorption on hydroxyapatite as a function of pH is shown for the initial concentration of tartrate ions 0.000001 mol/dm^3^ in Figure 5 and for the initial concentration of tartrate ions 0.001 mol/dm^3^ in Figure 6. As can be seen from Figure 3 at pH = 7.3, 77% of tartrate ions are adsorbed, and then, with an increase in pH, their adsorption drops to 39% at pH = 11.34. For an initial concentration of 0.001 mol/dm^3^ tartrate ions, Figure 6, at pH = 6.4, the adsorption is 79.2%; then, with an increase in pH = 7.15, it increases to 80.9%, and with a further increase in pH, there is a decrease in adsorption and in a solution with pH = 11.15 is 63%. The decrease in tartrate anion adsorption resembles anion adsorption at the metal oxide/electrolyte solution interface [34]; however, in the case of insoluble metal oxide, the mechanism of anion adsorption is associated with the reaction of forming surface complexes, as a result of which the anion adsorption with a decrease in pH increases, and at a sufficiently high pH, drops to approx. 0%. Adsorption of carboxylic acid anions at the hydroxyapatite/electrolyte solution interface, as mentioned above, is a more complicated process in which ion exchange processes depend not only on the concentration of H^+^/OH^−^ ions but also on the concentration of phosphate ions released as a result of dissolution of hydroxyapatite and adsorption of carboxylic ions [35]. The analysis of the composition of tartrate forms in the aqueous solution as a function of pH indicates that calcium tartrate complexes occur at concentrations resulting from the solubility of hydroxyapatite in the pH range corresponding to the occurrence of the tartrate complex, i.e., below pH = 6. Thus, in the studied pH range, tartrate ion interacts with hydroxyapatite. 

Based on the dependencies presented in Figure 5 and Figure 6 and the dependence of the adsorption of tartrate ions on hydroxyapatite from solutions with initial concentrations of 0.00001 and 0.0001 mol/dm^3^ of tartrate ions, the adsorption and concentration of tartrate ions for the three selected pH values, i.e., 7.5, 9.0 and 11.0 made it possible to present in Figure 7 the adsorption as a function of the concentration of tartrate ions. Figure 7 shows the concentrations of hydroxyl, phosphate and calcium ions groups on the surface, calculated from the data reported in the papers by Kukura et al., Skartsila and N. Spanos and Bertinetti et al. [36,37,38]. A comparison of the concentration of calcium ions, phosphate and hydroxyl groups in the case of adsorption of tartrate ions from a solution with an initial concentration of 0.001 mol/dm^3^ from a solution at pH = 7, indicates that the surface coverage in relation to these groups is 12% or 18% or 54%, respectively. It follows that, under the studied conditions, tartrate ion adsorption was lower for each of the adsorption centres than the formal monolayer coating. Analyzing the adsorption relationship presented in Figure 5 in the form of log (ion adsorption) as a function of log (equilibrium concentration) for the tested pH, the presented relationship has a linear course with a slope of ~1, which indicates that ion adsorption occurs on a surface with energetically homogeneous sites [39]. However, the fit of the Langmuir isotherms to the obtained data in Figure 7 is worse than that of Freundlich isotherms (Table 1). Similar results of matching both isotherms were obtained for adsorption of oxalic, malic and citric acids on hydroxyapatite [14].
(3)$$a_{c}=\frac{a_{m}K_{L}c_{e}}{1+K_{L}C_{e}}$$
where*a_c_* (mol/m^2^)—the adsorption of tartrate ions.*C_e_*—the equilibrium concentration of tartrate ions.*a_m_*—the maximum amount of tartrate ions.*K_L_*—the constant related to the binding energy.(4)ac=KFce1n
where *K_F_* and 1/*n* are the empirical constants of Freundlich isotherm.

**Figure 7 materials-14-03039-f007:**
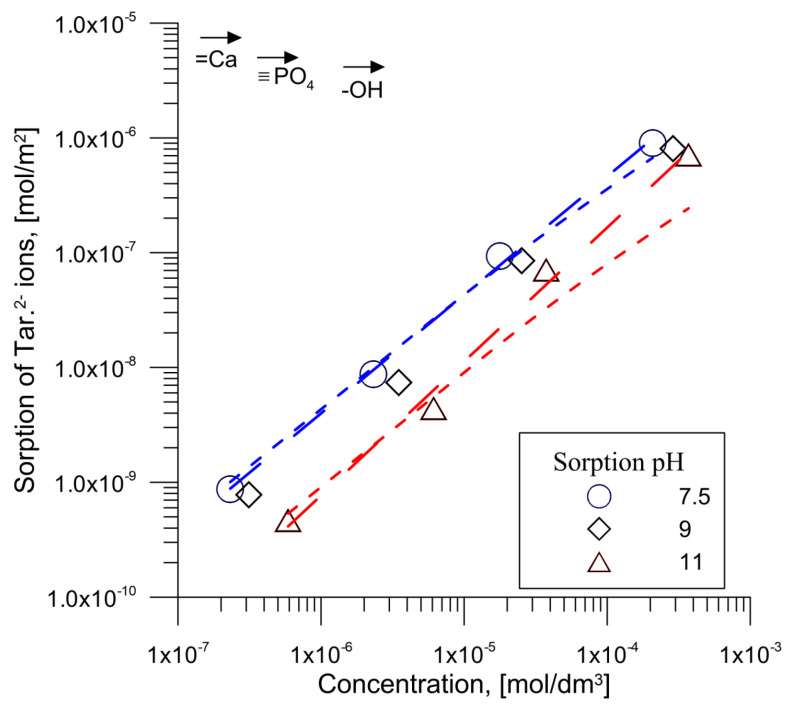
Dependence of log (adsorption density) as a function of log (concentration of tartrate ions) for selected pH for the hydroxyapatite/0.001 mol/dm^3^ NaCl + 0.000001 mol/dm^3^ tartrate ions. The short dash line indicates the Langmuir isotherm fit, and the long dash line points out to the Freundlich isotherm fit.

**Table 1 materials-14-03039-t001:** Parameters of Langmuir and Freundlich isotherms for the adsorption of tartrate ions on the hydroxyapatite surface.

**Parameters**	**pH**		
	7.5	9	11
**Parameters of Langmuir Isotherm**			
a_m_	2.62 × 10^−6^	1.63 × 10^−6^	8.75 × 10^−7^
K_L_	1653.7	1653.7	1039.2
R^2^	0.90	0.77	0.21
**Parameters of Freundlich Isotherm**			
K_F_	0.0062	0.0039	0.0077
n	0.969	0.968	0.857
R^2^	0.99	0.99	0.99

R^2^—Coefficient of determination [40].

Figure 8 presents the amount of H^+^ ions released as a result of adsorption of tartrate ions on hydroxyapatite as a function of pH. As can be seen, starting with an initial concentration of 0.0001 mol/dm^3^ tartrate ions, their adsorption releases H^+^ ions simultaneously, which can indicate that acidic groups appear on the surface of hydroxyapatite. The precise determination of their amount based on titration is unreliable due to the solubility of hydroxyapatite affected by carboxylic acids [14]. Nevertheless, due to the fact that in the second dissociation level tartaric acid is stronger than hydrogen phosphoric acid and hydroxyl groups, the reaction of exchanging these groups for tartrate anion may increase the acidic nature of surface groups due to the of tartrate ions adsorption. This effect is confirmed by measurements of the zeta potential. In the presence of tartrate ions with the initial concentrations of 0.0001 and 0.001 mol/dm^3^ in the hydroxyapatite system, the electrolyte solution is followed by a decrease in the zeta potential (Figure 9) caused by the increase of negatively charged groups on the surface of hydroxyapatite due to the adsorption of tartrate ions.

### 3.2. FTIR Study of Tartrate Ions Adsorption on Hydroxyapatite

The analysis of the location of the characteristic bands presented in the substracted spectrum confirms the presence of the bands characteristic of calcium tartrate for the wave numbers 1122, 1083, 1047, 1008, 962, 602 and 559 cm^−1^ [41]. In the FTIR spectrum, the values of wavenumbers 1445 and 1413 cm^−1^ were characteristic of hydroxyapatite substitution with carbonate ions as a result of CO_2_ adsorption during the tartaric acid adsorption study on hydroxyapatite. Grunenwald et al. developed a method for determining the content of carbonate ions in hydroxyapatite samples using FTIR spectroscopy [42]. In order to determine the percentage of carbonate, the area of the vibration-characteristic ν_1_ and ν_3_ phosphate ions peaks is calculated in the range of wave numbers 900 to 1230 cm^−1^ and the area of peaks characteristic for vibrations ν_3_ of carbonate ions in the range of wave number 1330–1550 cm^−1^; Figure 10. After calculating the area under the peaks for the samples, HAp conditioned in 0.001 mol/dm^3^ NaCl solution and HAp sample conditioned in 0.001 mol/dm^3^ NaCl + 0.001 mol/dm^3^ Tart, using the Grunenwald et al. formula, the percentage of carbonate ions in these samples was calculated. The HAp sample conditioned in 0.001 mol/dm^3^ NaCl + 0.001 mol/dm^3^ contained 1.6% by weight of carbonate ions, while the conditioned sample in 0.001 mol/dm^3^ NaCl with an addition of 0.001 mol/dm^3^ of tartaric acid contained 1.4% by weight. The lower content of carbonates in the conditioned sample in the solution of 0.001 mol/dm^3^ NaCl + 0.001 mol/dm^3^ of tartaric acid can be attributed to the reaction of carbonate exchange with tartrate anions. To compare the loss of carbonate ions, this mechanism has a ~37% share in the adsorption of tartrate ions.

### 3.3. PXRD Study of Hydroxyapatite Samples

The comparison of the crystalline structure of hydroxyapatite samples before and after the adsorption of tartrate ions on the basis of PXRD patterns was made by the Rietveld method using the Maud program [43]. The examples of the results of adjusting the diffraction data and calculations by the Rietveld method are presented in Figure 11. The lattice constants of the cell and the fitting parameters are listed in Table 2. The obtained values of lattice constants are in good agreement with the corresponding constants PDF cards 00-009-0432. As can be seen, as a result of conditioning the hydroxyapatite sample for 14 days in 0.001 mol/dm^3^ NaCl + 0.001 mol/dm^3^ tartrate solution, no new phase is formed, only lattice constants are slightly reduced due to the adsorption of tartrate ions. This is evident in the column relating to the volume of the elemental cell.

Ion substitution in the crystal lattice can affect not only the solid lattice of the unit cell but also the crystal size and crystallite index. The crystallite index, CIXRD, was calculated on the basis of the height of reflections (202), (300) (112) and (211) by the Person et al. method [44], while the crystal size was determined by the Scherrer method for reflections (002) and (003) [45]. The obtained calculation results for reflections (002) do not change; however, for reflections (003) of the sample conditioned for two weeks in a solution of 0.001 mol/dm^3^ of tartrate ions, it decreases significantly, which may suggest crystal dissolution (Table 3). Similarly, for this sample, the crystallite index decreases. 

### 3.4. Study of the Surface Area, Porosity and Particle Size

The nitrogen desorption–adsorption method was used for determination of the effect of hydroxyapatite conditioning for 14 days in 0.001 mol/dm^3^ NaCl, in a solution in 0.001 mol/dm^3^ NaCl + 0.0001 mol/dm^3^ tartrate ions and in a solution containing 0.001 mol/dm^3^ NaCl + 0.0001 mol/dm^3^ tartrate ions specific surface area and porosity. The measurement results are given in Table 4, and the distribution of pores in Figure 12. As can be seen from the results, the samples conditioned in the electrolyte solutions (NaCl, NaCl + tartrate ions) for 14 days have a smaller specific surface area; compared to that of the original sample, the remaining samples have a larger pore size; pore distribution dA vs. d(log (D)) indicates that conditioning leads to a reduction in the surface area associated with the 2 to 20 nm diameter pores. However, the differences in the distribution of samples conditioned in the presence of tartaric acid are small. This effect may be caused by the clogging of small pores by the electrolyte remaining after conditioning, which is difficult to remove from pores with a small diameter.

The results of particle size analysis of the primary sample of hydroxyapatite and in the samples conditioned for 14 days in 0.001 mol/dm^3^ NaCl, in a solution with a composition 0.001 mol/dm^3^ NaCl + 0.0001 mol/dm^3^ tartrate ions and in a solution containing 0.001 mol/dm^3^ NaCl + 0.0001 mol/dm^3^ tartrate ions are shown in Figure 13. The particle size distribution of the sample conditioned in 0.001 mol/dm^3^ NaCl solution coincides with the particle size distribution of the primary sample (Initial). In the case of the sample conditioned in 0.0001 mol/dm^3^ tartrate solution, there was a shift observed in the numerical particle size distribution towards larger particle diameters. This effect may be caused by an increase in the electrolyte ionic strength and aggregation of particles. On the other hand, in the solution containing 0.001 mol/dm^3^ of tartrate ions, the effect is opposite; there is a shift towards smaller diameters which may be associated with the dispersion of aggregates affected by ultrasounds and a decrease in the value of the zeta potential (Figure 9).

## 4. Conclusions

The studies on the kinetics of tartrate ion adsorption from a solution with an initial concentration of 0.000001 mol/dm^3^ indicate that the equilibrium of adsorption is established within 10 min, ~89% of the initial concentration of tartaric acid was adsorbed. However, the kinetics of tartaric acid adsorption from a solution with a concentration of the initial 0.001 mol/dm3 is more complex solution and is best described using a multiexponational model, which proves successive processes with a variable adsorption mechanism, i.e., adsorption/ion exchange on the surface, and later ion diffusion inside type II channels.

Adsorption of tartrate ions decreases with the increasing pH for all tested initial concentrations. This may indicate the exchange of hydroxyl or/and phosphate ions with tartrate ions. This process leads to an increase in the concentration of groups with a negative charge on the surface of hydroxyapatite, which is confirmed by the amount of H^+^ ions released as a result of tartrate adsorption and a decrease in the zeta potential.

The size of the crystals calculated on the basis of reflections (003) of a sample conditioned for two weeks in a solution of 0.001 mol/dm^3^ of tartrate ions decreases significantly, which may suggest the dissolution of the crystal and the shift of the grain distribution towards smaller diameters, as shown by the dissolution of hydroxyapatite crystals.

Research on the adsorption of tartaric acid and hydrosypapaptite may be helpful in improving the conditions for the preparation of the enamel surface for the application of polymers in dentistry.

## Figures and Tables

**Figure 1 materials-14-03039-f001:**
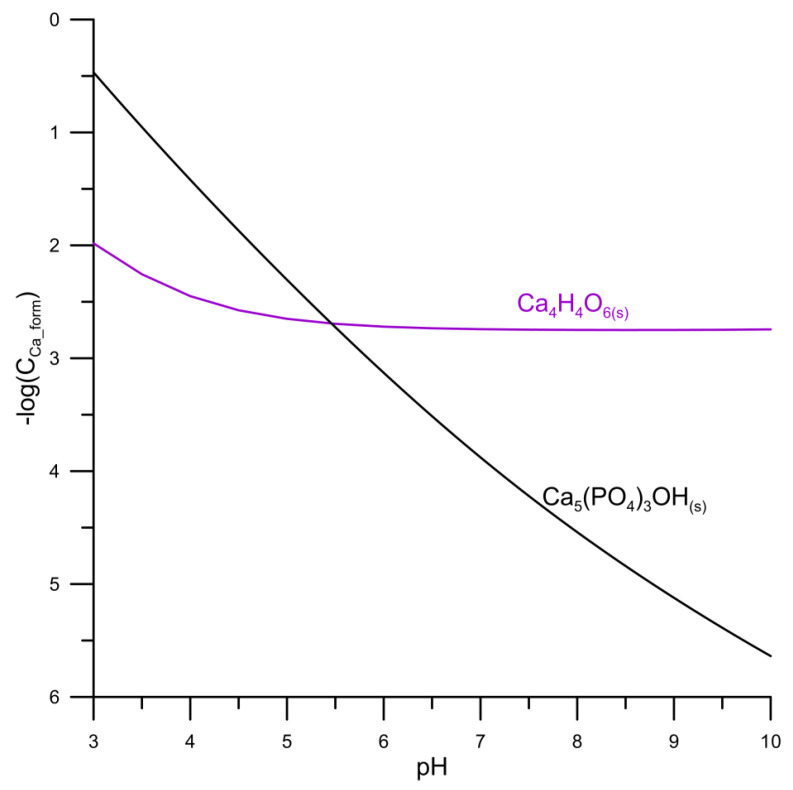
The relationship of the negative logarithm of the sum of concentrations of calcium species in the hydroxyapatite/water system and calcium tartrate/water.

**Figure 2 materials-14-03039-f002:**
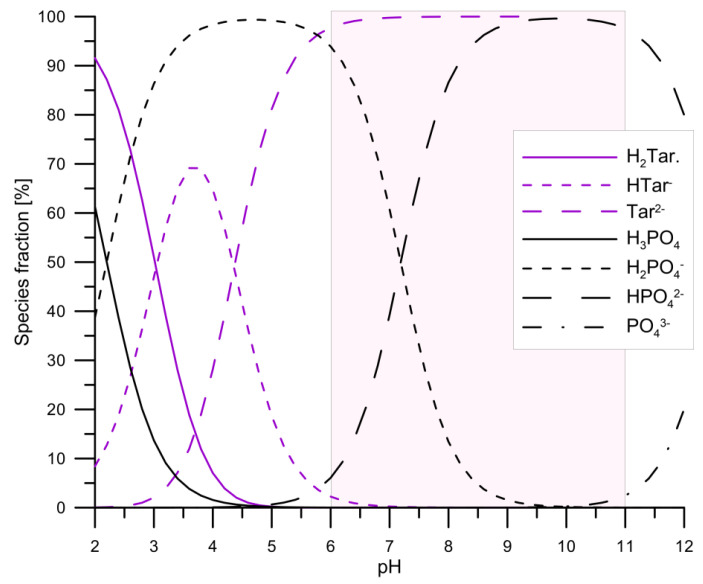
The comparison of tartrate and phosphate species as a function of pH in aqueous solutions. The rectangle denotes experimental study the pH range.

**Figure 3 materials-14-03039-f003:**
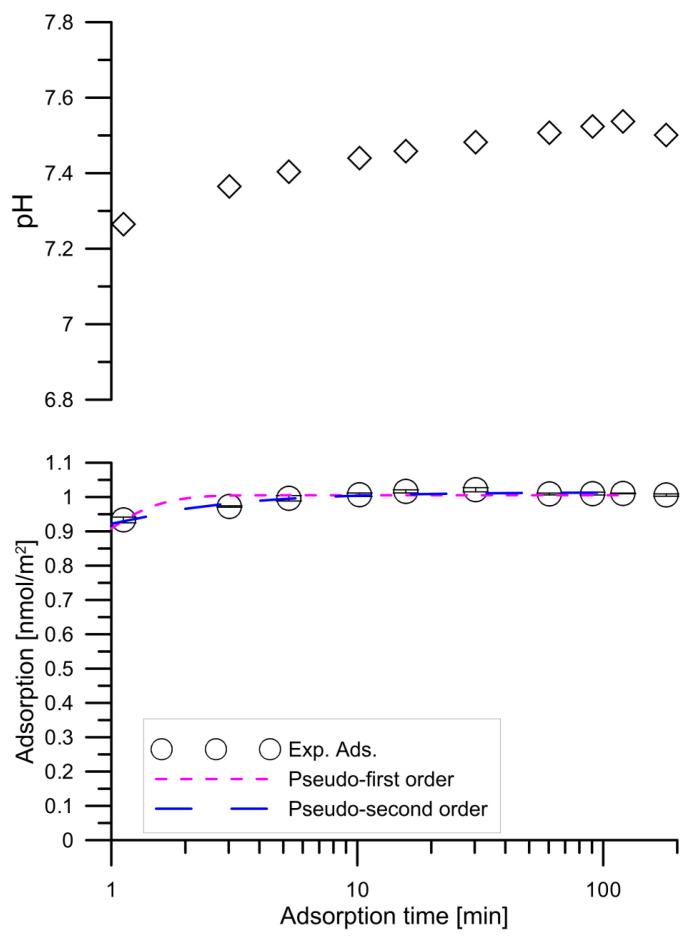
Changes of kinetics of pH and tartrate ions adsorption on hydroxyapatite from the tartrate ions solution of the initial concentration 0.000001 mol/dm^3^. The open circle points indicate the experimental adsorption data; open diamond indicates pH; the dashed line of short segments indicates the pseudo-first order model for *q_e_* = 1.005 nmol/m^2^ and *k*_1_ = 2.34; the dashed line of longer segments indicates the pseudo-second order model for *q_e_* = 1.014 nmol/m^2^ and *k*_2_ = 9.92.

**Figure 4 materials-14-03039-f004:**
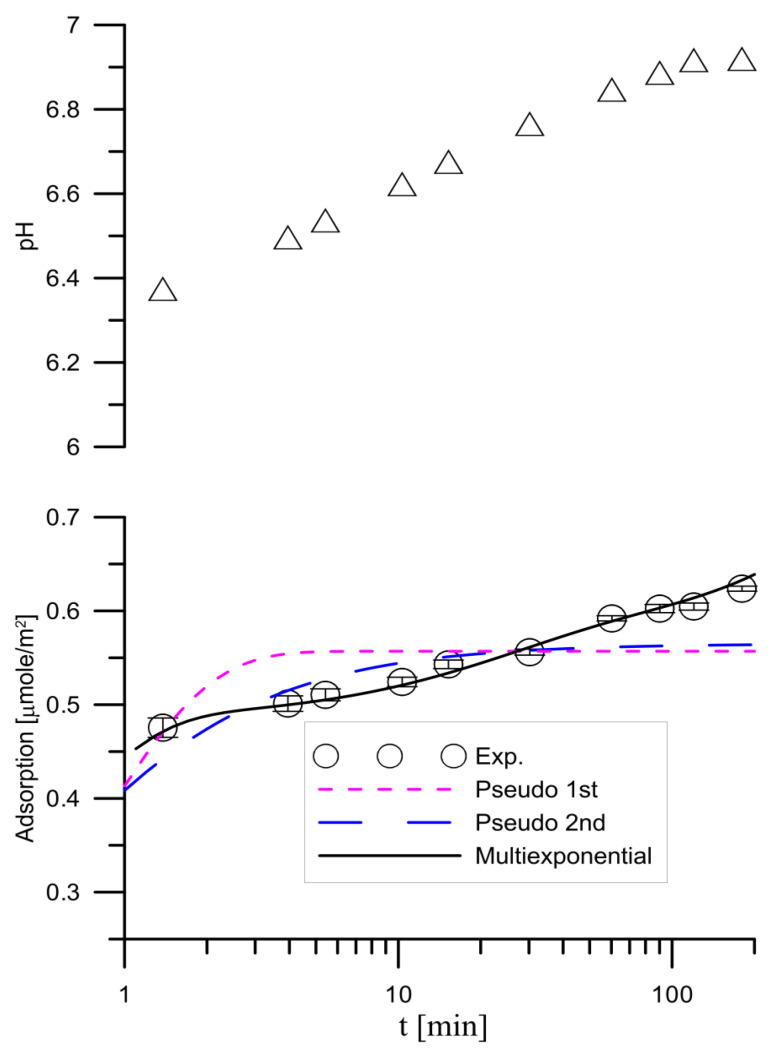
Changes of kinetics of pH and tartrate ions adsorption changes on hydroxyapatite from the tartrate ions solution of the initial concentration 0.001 mol/dm^3^. The open circle indicates the experimental adsorption data; the dashed line with short segments indicates the pseudo-first order model for *q_e_* = 0.56 μmol/m^2^ and *k*_1_ = 1.35; the dashed line of longer segments indicates the pseudo-second order model for *q_e_* = 0.58 and *k*_2_ = 4.63. The continuous line indicates the multiexponential model: *a_eq_* = 1.402, *A*_0_ = 0.405, *k*_1_ = 2.38, *A*_1_ = 0.207, *k*_2_ = 2.97 × 10^−4^, *A*_2_ = 0.348, *k*_3_ = 0.045, *A*_3_ = 0.039.

**Figure 5 materials-14-03039-f005:**
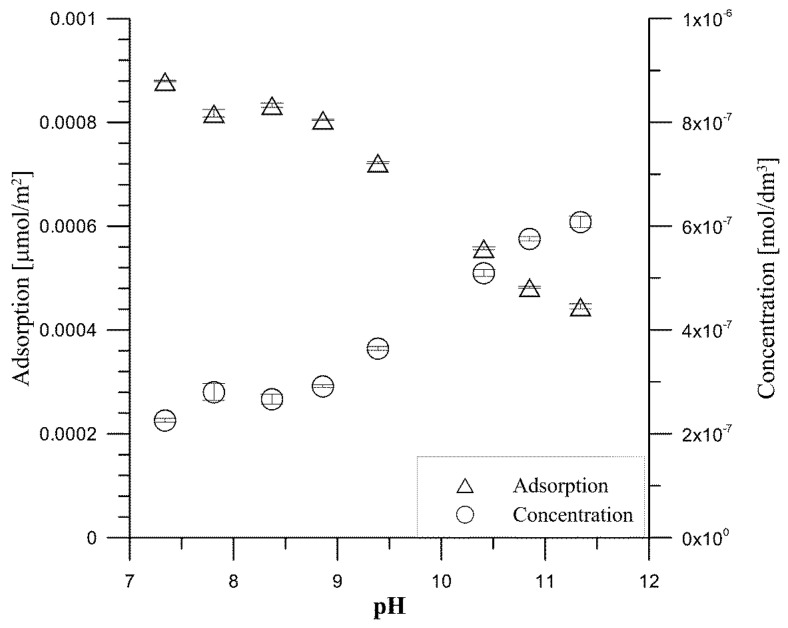
Adsorption density and concentration of tartrate ions as a function of pH for the hydroxyapatite/0.001 mol/dm^3^ NaCl + 0.000001 mol/dm^3^ tartrate ions.

**Figure 6 materials-14-03039-f006:**
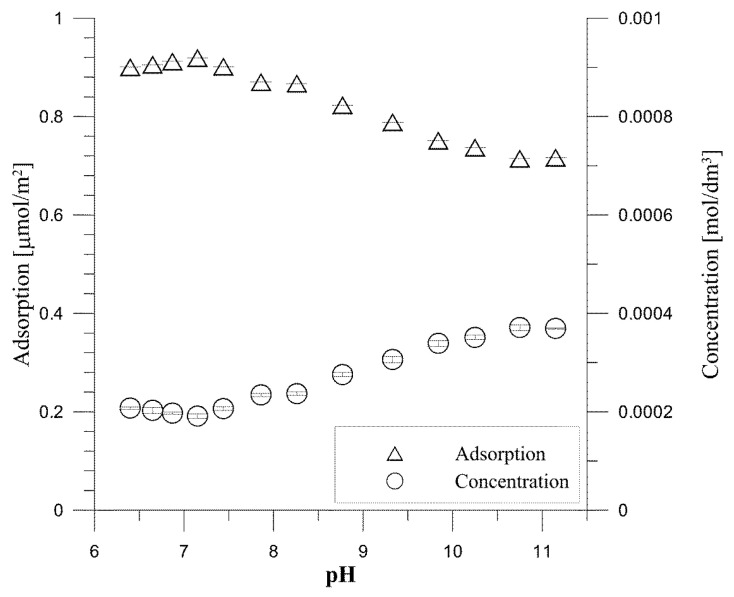
Adsorption density and concentration of tartrate ions as a function of pH for the hydroxyapatite/0.001 mol/dm^3^ NaCl + 0.001 mol/dm^3^ tartrate ions.

**Figure 8 materials-14-03039-f008:**
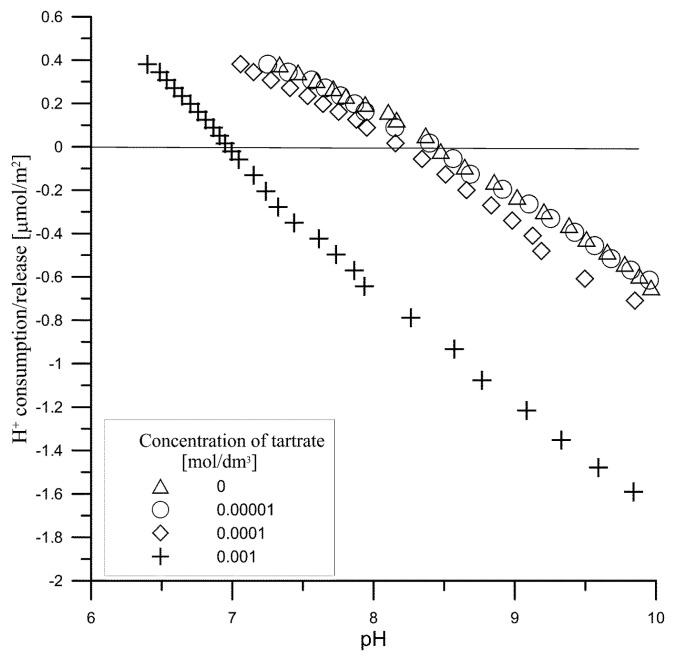
The consumption/release of H^+^ ions upon tartrate ions adsorption in the hydroxyapatite/0.001 mol/dm^3^ NaCl solution.

**Figure 9 materials-14-03039-f009:**
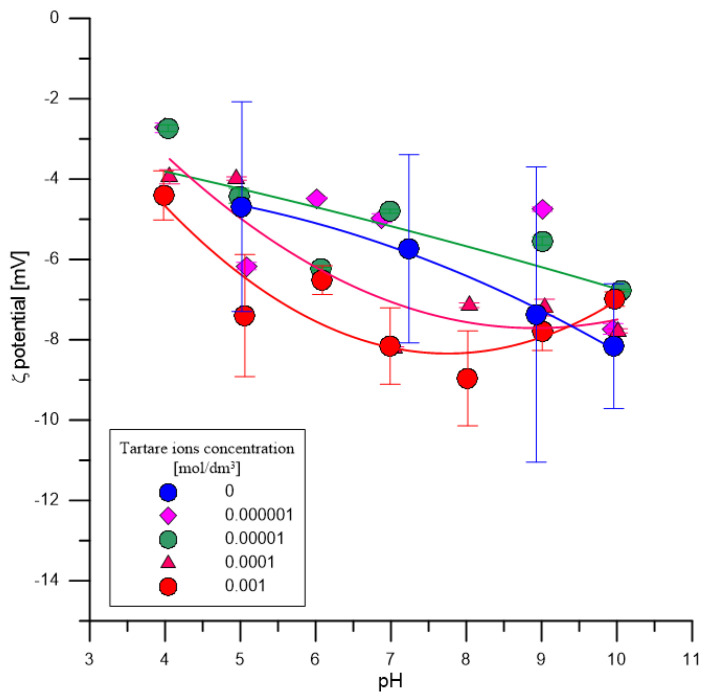
Dependence of zeta potential as a function of pH for the effect of tartrate adsorption on hydroxyapatite.

**Figure 10 materials-14-03039-f010:**
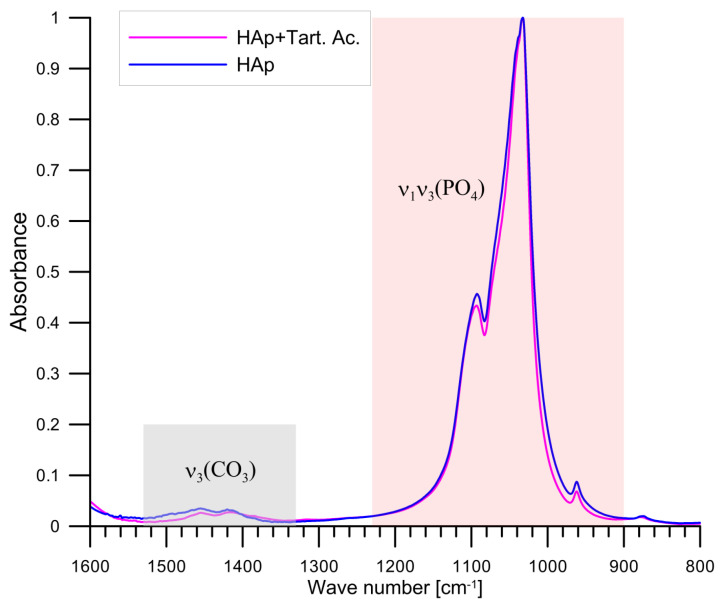
Comparison of FTIR spectra of sample of hydroxyapatite conditioned for 14 days in 0.001 mol/dm^3^ NaCl with sample conditioned in a solution with a composition 0.001 mol/dm^3^ NaCl + 0.0001 mol/dm^3^ tartrate ions.

**Figure 11 materials-14-03039-f011:**
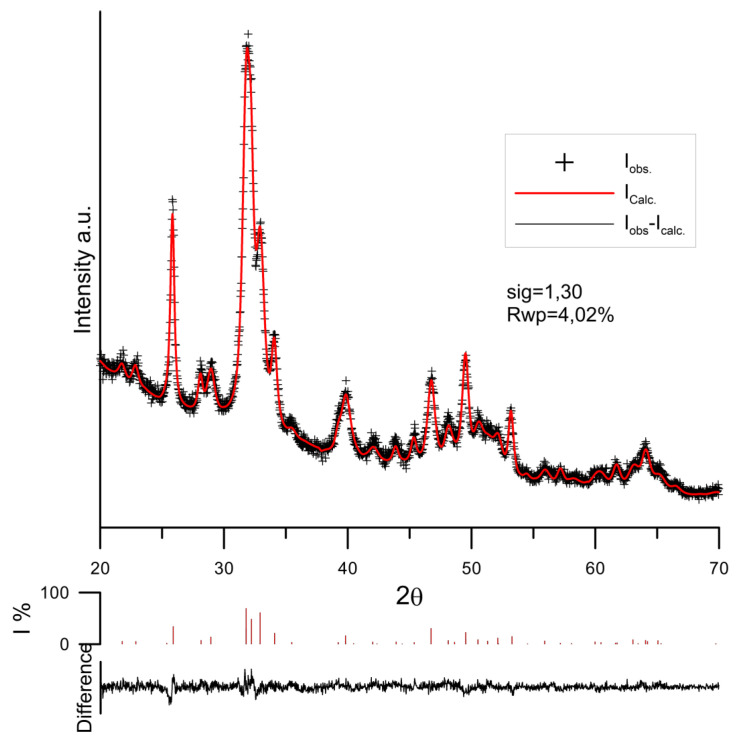
PXRD pattern of hydroxyapatite with adsorbed tartrate ions after 2 weeks of conditioning in the solution of the initial concentration 0.001 mol/dm^3^ of tartrate ions.

**Figure 12 materials-14-03039-f012:**
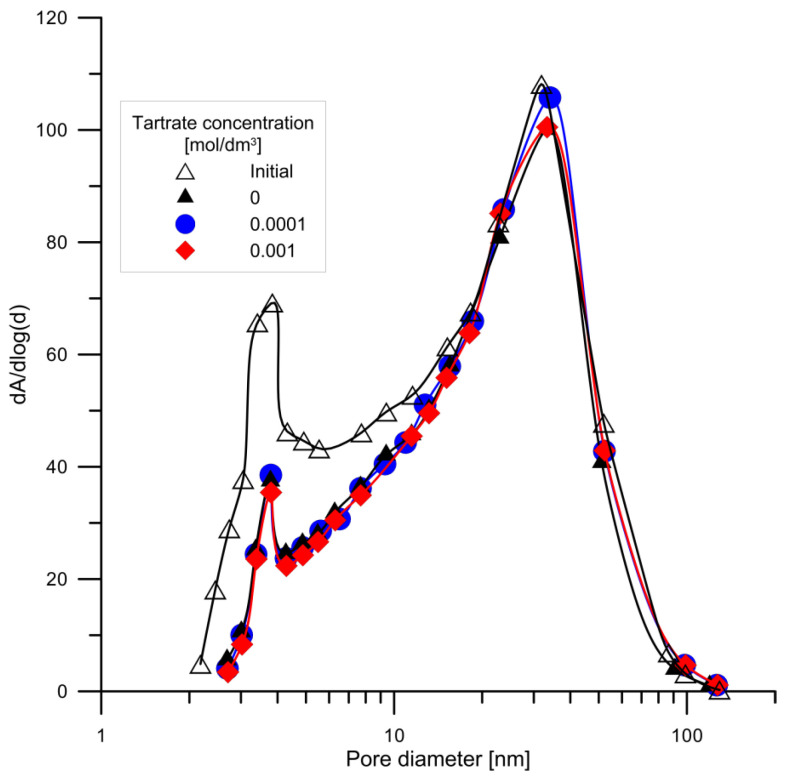
Comparison of pore distribution in the original sample of hydroxyapatite and in the samples conditioned for 14 days in 0.001 mol/dm^3^ NaCl, in a solution with a composition 0.001 mol/dm^3^ NaCl + 0.0001 mol/dm^3^ tartrate ions.

**Figure 13 materials-14-03039-f013:**
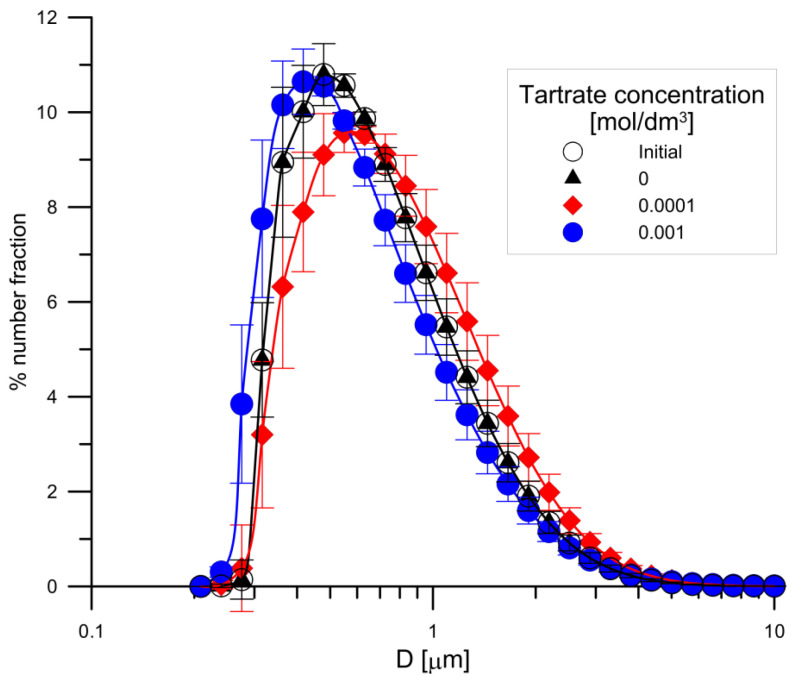
Comparison of numerical particle size distribution of the initial sample of hydroxyapatite (initial) and in the samples conditioned for 14 days in a solution of 0.001 mol/dm^3^ NaCl, in a solution with a composition in 0.001 mol/dm^3^ NaCl + 0.0001 tartrate ions and in a solution containing 0.001 mol/dm^3^ NaCl + 0.0001 tartrate ions.

**Table 2 materials-14-03039-t002:** The hydroxyapatite lattice constants.

Sample Code	a [Å]	c [Å]	V [Å^3^]	Sig	Rwp [%]
HAp/0.001 mol/dm^3^ NaCl	9.41392	6.8886	528.69	0.2	5.52
HAp/0.0001 mol/dm^3^ Tart.	9.41043	6.8848	528.01	1.86	5.78
HAp/0.001 mol/dm^3^ Tart.	9.40813	6.8850	527.77	1.30	4.02
PDF card no 00-009-0432	9.4180	6.8840	528.80	-	-

**Table 3 materials-14-03039-t003:** The crystal size and crystality index of the HAp samples conditioned for 2 weeks in the aqueous electrolyte solutions.

Sample	D(002) [nm]	D(300) [nm]	D(002)/D(300)	CI_XRD_ [46]
HAp/0.001 mol/dm^3^ NaCl	26.5 ± 0.3	14.0 ± 0.3	1.89	10.19
HAp/0.0001 mol/dm^3^ Tart.	26.5 ± 0.4	14.1 ± 0.4	1.89	9.95
HAp/0.001 mol/dm^3^ Tart.	26.6 ± 0.5	13.4 ± 0.3	1.99	9.86

**Table 4 materials-14-03039-t004:** Parameters characterizing the specific surface area and porosity of hydroxyapatite samples.

Property	Sample			
	HAP (Initial)	HAP/0.001 mol/dm^3^ NaCl	HAP/0.0001 mol/dm^3^ Tartr.	HAP/0.001 mol/dm^3^ Tartr.
BET Surface Area [m^2^/g]	87.7	69.6	73.7	73.6
Langmuir Surface Area [m^2^/g]	128.2	101.9	107.8	107.5
BJH Adsorption cumulative volume of pores 1.7 nm < d < 300 nm diameter [cm^3^/g]	0.53	0.46	0.51	0.50
BJH Desorption cumulative volume of pores 1.7 nm < d <300 nm diameter [cm^3^/g]	0.53	0.47	0.51	0.51
Adsorption average pore width (4 V/A by BET): [nm]	24.2	26.7	27.9	27.6
BJH Adsorption average pore diameter (4 V/A): [nm]	24.0	27.2	28.6	28.4
BJH Desorption average pore diameter (4 V/A): [nm]	22.6	24.8	26.0	26.2

## Data Availability

The data underlying this article will be shared on reasonable request from the corresponding author.

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
