# Peer review of "Adsorption of the Tartrate Ions in the Hydroxyapatite/Aqueous Solution of NaCl System"

_materials, 2021, doi:10.3390/ma14113039_

Round 1

Reviewer 1 Report

General comment

" Adsorption of the tartrate ions in the hydroxyapatite/aqueous 2 solution of NaCl system"

It is very interesting that the kinetics and static adsorption of tartrate ions on hydroxyapatite was investigated using 14C radiolabeled tartaric acid from solutions of tartaric acid at different concentrations and pH ranges. The manuscript is well written. However, there are a few corrections that are essential to meet the standard for publication. Please refer to the following comments.

1) This your research paper is a study of kinetics and static adsorption of tartrate ions on hydroxyapatite. What fields will the results of your research be useful in, such as clinical and preventive medicine? Please add your opinion on the outlook and prospect.

2) Many of the citations in your paper are old. Please cite as many newer, updated papers as possible.

Author Response

Review 1

Thank you very much for your comments, we marked all changes in the publication in yellow.

1) This your research paper is a study of kinetics and static adsorption of tartrate ions on hydroxyapatite. What fields will the results of your research be useful in, such as clinical and preventive medicine? Please add your opinion on the outlook and prospect.

The conclusions of the study include information on the possible use of the test results in the preparation of tooth surfaces in dentistry with the use of tartaric acid.

Unlike the interaction of other carboxylic acids with hydroxyapatite, the literature describing the interaction of tartaric acid with hydroxyapatite is sparse and published papers deal with the decalcification of tooth enamel. We found no papers on the adsorption of tartaric acid / tartrate ion on hydroxyapatite.

2) Many of the citations in your paper are old. Please cite as many newer, updated papers as possible.

We have introduced the following new citations.

  1. Janusz and E. Skwarek, Comparison of Oxalate, Citrate and Tartrate Ions, Adsorption in the Hydroxyapatite/Aqueous Electrolyte Solution System. Colloids and Interfaces, 4(2020)45; doi:10.3390/colloids4040045

[1] A. Grunenwald, C. Keyser, A.M. Sautereau, E. Crubézy, B. Ludes, C. Drouet Revisiting carbonate quantification in apatite (bio)minerals: a validated FTIR methodology. Journal of Archaeological Science, 49 (2014) 134-141.

Reviewer 2 Report

After reading the paper I have following remarks.

  1. The abstract should be rewritten. The Authors should cleary, briefly and in the following order described (i) an research subject, (ii) what data were used, (iii) what methods were used, (iv) what results were obtained and (v) the most important conclusion.
  2.  Why did the Authors choose the initial concentration of tartaric acid  from 0.000001 mol/dm3 to 0.001 mol/dm3 and the pH range from 6 to 11 and the 0.001 mol/dm3 aqueous solution of NaCl. It should be clearly justified.
  3.  The Authors used two types of water i.e. double destilled water and deionized water. For what reason? What was a electrical conductivity of deionized water? It should be explained and corrected.
  4.  In Table 1, "Langmuir", "Freundlich" should be changed. I suggest "Parameters of the Freundlich isoterm" etc.
  5. What was the accuracy of the pH-value measurement? 9.5 +/-0.1 oraz 9 +/-1? (see Table 1). It should be corrected?
  6. In my opinion, section "conclusions" is written too extensively. These are supposed to be conclusions, not a study summary. It should be rewritten.
  7. What are the practical conclusions/applications/possible use the research findings/methods etc.

Author Response

Review 2

Thank you very much for your comments, we marked all changes in the publication in yellow.

1.The abstract should be rewritten. The Authors should cleary, briefly and in the following order described (i) an research subject, (ii) what data were used, (iii) what methods were used, (iv) what results were obtained and (v) the most important conclusion.

The abstract has been corrected based on the above-mentioned guidelines.

2.Why did the Authors choose the initial concentration of tartaric acid  from 0.000001 mol/dm3 to 0.001 mol/dm3 and the pH range from 6 to 11 and the 0.001 mol/dm3 aqueous solution of NaCl. It should be clearly justified.

In the manuscript, line 150, we wrote that the range of measurements was limited from pH = 6 to 11 due to the solubility of hydroxyapatite, Fig 1. At pH = 6 due to solubility and not taking into account decalcification, in a solution of 0.05 dm3, the sample reduces its mass by ~ 3%, below this pH the solubility increases sharply and precipitation of calcium tartrate may occur. The adsorption measurements from the solution with an initial concentration of 1*10-6 mol/dm3 were caused by the specific radioactivity of 14C-labeled tartaric acid, which ensured that a sample from a solution with a volume of 0.1 cm3 before adsorption would provide 20,000 imp/min in the scintillation counter, while from the adsorption solution to measure the radioactivity will be many times different from the background. These conditions will allow an accurate measurement of the adsorption. No measurements were performed at higher than 0.001 mol/dm3 of tartaric acid to avoid excessive dissolution of hydroxyapatite.

3.The Authors used two types of water i.e. double destilled water and deionized water. For what reason? What was a electrical conductivity of deionized water? It should be explained and corrected.

We apologize for the mistake in the given values of the water conductivity after washing the hydroxyapatite, it should be 10.7 ms/cm. In the description of the purification of hydroxyapatite, we wrote that we used double-distilled water, which has a conductivity of 2mS/cm and is pure enough for washing the hydroskyaptite, while Mili-Q water with a conductivity of 0.5mS/cm was used to prepare the solutions.

  1. In Table 1, "Langmuir", "Freundlich" should be changed. I suggest "Parameters of the Freundlich isoterm" etc.

We corrected the headings in Table 1.

5.What was the accuracy of the pH-value measurement? 9.5 +/-0.1 oraz 9 +/-1? (see Table 1). It should be corrected?

The data for the preparation of the dependence of adsorption as a function of the equilibrium concentration of tartrate ions was obtained by adjusting the adsorption dependence as a function of pH with a 4th degree polynomial, and then using factors in polynomials, the adsorption values for selected pHs, i.e. 7.5, 9 and 11, were calculated. The manuscript contains information on the obtained adsorption values and tartaric acid concentration for pH = 7.5, 9 and 11.

6.In my opinion, section "conclusions" is written too extensively. These are supposed to be conclusions, not a study summary. It should be rewritten.

The conclusions were corrected in line with the above-mentioned suggestions.

7.What are the practical conclusions/applications/possible use the research findings/methods etc.

 The conclusios contains information on the possible use of the test results in the preparation of the tooth surface in dentistry with the use of tartaric acid.